# Improvement in Migration Resistance of Hydroxyl-Terminated Polybutadiene (HTPB) Liners by Using Graphene Barriers

**DOI:** 10.3390/polym14235213

**Published:** 2022-11-30

**Authors:** Yanan Zhang, Yu Tian, Yulong Zhang, Xuelong Fu, Hao Li, Zhehong Lu, Tengyue Zhang, Yubing Hu

**Affiliations:** 1College of Materials Science and Engineering, Nanjing Tech University, Nanjing 211816, China; 2China North Industry Advanced Technology Generalization Institute, Beijing 100089, China; 3Department of Mechanical & Electronic Engineering, Jiangsu Polytechnic of Finance & Economics, Huai’an 223003, China; 4National Special Superfine Powder Engineering Research Center of China, Nanjing University of Science and Technology, Nanjing 210014, China; 5Ordnance Science and Research Academy of China, Beijing 100089, China

**Keywords:** composite liner, modified GO, anti-migration, bonding properties

## Abstract

The excessive migration of plasticizers leads to debonding and cracking of a liner, which can compromise the safety of a solid propellant. Graphene oxide (GO), with a laminar structure as a filler, can effectively reduce the migration of plasticizers. In this study, we modified GO using toluene diisocyanate (TDI). The cross-link density of the substrate was increased by grafting isocyanate groups to obtain a denser liner for the purpose of preventing plasticizer migration. We also used octadecylamine (ODA) to modify GO by grafting negatively charged amide groups on the GO surface. The electrostatic repulsive effect of the amide group on the plasticizer molecules was used to prevent plasticizer migration. Two modified GOs were filled into the hydroxyl-terminated polybutadiene to prepare two composite liners. We then investigated the migration resistance and migration kinetics of each modified liner using the dipping method. In addition, we explored the mechanical properties of each modified liner. Compared with the original liner, the anti-migration and mechanical properties of the modified composite liners were significantly improved. Among them, the TDI-modified liner had the most obvious improvement in migration resistance, while the ODA-modified liner had the greatest improvement in bonding properties. All types of liners met the requirements of the current propellant systems. This study provides an effective reference for improving the migration resistance and bonding properties of the composite liner.

## 1. Introduction

During the long-term storage of propellant, the interface between the propellant and the adiabatic layer will gradually fail due to the migration of the components. If the propellant is used at this time, interface debonding can occur by high speed, high pressure and high temperature gas, and eventually develop into cracks or voids and other defects, affecting the overall performance and storage life of the solid rocket motor [1]. In solid rocket motors, to stop the excessive migration of small molecules such as propellant, it is necessary to use a liner to protect the junction between the propellant and the adiabatic layer [2]. Hydroxyl-terminated polybutadiene (HTPB) is a claw-shaped liquid prepolymer, which can be cured and cross-linked at room temperature or high temperature to form an elastomer with a three-dimensional network structure. It has many excellent physical properties, such as low viscosity, low glass transition temperature, high thermal stability and good corrosion resistance. Therefore, HTPB is often used as a liner for solid propellants [3,4,5].

In the propellant/liner/insulation layer interface system, the plasticizer migrates from the grain to the insulation layer due to the concentration difference of the plasticizers [6,7,8]. The migration of the plasticizer will weaken the interfacial bonding between the systems [9]. In addition, the migration of the plasticizer from the propellant leads to the shrinkage of the propellant, and the migration of the plasticizer into the insulation layer causes the insulation layer to swell, which ultimately destroys the safety and stability of the engine [10]. However, there are few reports on the inhibition of plasticizer migration. The preparation of composite materials by adding fillers with various properties to the matrix material is the most common and effective method [11,12,13]. In order to inhibit the migration of plasticizers, researchers have adopted many methods, including the modification of the liner materials and the addition of inorganic fillers [14,15]. In general, polymer matrix with low polarity, high crystallinity and cross-link density has a low passage rate for plasticizer small molecules. Therefore, the liner can be modified in terms of enhancing the cross-link density of the liner to make it more resistant to migration. Wu et al. [16] prepared a liner with high cross-link density and low polarity by mixing 3-aminopropyltriethoxysilane, phenolic resin and polyvinyl butyral. They designed a three-layer barrier liner between the propellant and the insulation layer. The migration of the plasticizer was effectively reduced. Adding inorganic nanofillers to the liner composites is one of the effective methods to inhibit the migration of plasticizers. Farajpour et al. [17] reduced the migration of dioctyl phthalate plasticizers from polyurethane to ethylene propylene diene monomer substrates by dispersing carbon nanosheets in polyurethane liners. Wu et al. [18] added layers of silicate to the ethylene propylene diene monomer/neoprene propellant liner to block the migration of triacetin and nitrocellulose. According to the polar interaction hypothesis, many polymers are able to provide electrons, while most of the plasticizer molecules are able to accept them. The plasticizer molecules migrate due to this electrostatic attraction. Therefore, the introduction of negatively charged groups in the liner to form an electrostatic repulsion with the plasticizer molecules is also a way to enhance the migration resistance of the liner.

Graphene oxide, a novel planar nanomaterial, was widely used in nanocomposites because of its outstanding properties such as high surface area, high electrical conductivity, high thermal conductivity and superior mechanical properties [19,20,21,22,23]. In our previous study [24], we prepared composite liners with GO as a filler and systematically investigated the migration resistance of the liners. Graphene oxide forms a physical barrier through its unique layered structure, which weakens the adsorption capacity between small molecules by reducing the formation of ester groups. We also modified GO with toluene diisocyanate to prepare structurally modified GO/HTPB liners [25]. By comparison, it was found that the anti-migration performance of the modified liner was improved.

In solid rocket motors, the bonding interface of the propellant system is prone to interface debonding after being influenced by the outside world. Interfacial debonding will cause the propellant grain to burn out of control and cause an accident [2,26,27]. In order to avoid interfacial debonding, the liner is required to have certain mechanical properties, such as tensile properties and bonding properties. Zhang et al. [28] used poly (adipic diethylene glycol), glycerol and toluene diisocyanate to synthesize a polyester-based transition layer, which was applied between the propellant and the liner layer. This substantially improved the bonding properties between the interfaces and effectively inhibited the migration of high-energy plasticizers.

In this study, we used GO filled into HTPB to prepare the composite liner layer. Subsequently, we modified the GO using two separate methods with the aim of further enhancing the migration resistance and bonding properties of the liner. We used TDI to modify the liner with the aim of enhancing the cross-link density of the liner matrix and thus the migration resistance. Similarly, we also modified GO with ODA, hoping to introduce negatively charged amide groups into the liner matrix to enhance the migration resistance of the liner by using its electrostatic repulsion with the plasticizer molecules. As a comparison, we prepared four different liner layers, namely a pure HTPB liner, a GO/HTPB (GH) liner, a TDI-modified GO/HTPB (TGH) liner and an ODA-modified GO/HTPB (GAH) liner. The effects of the different GO modification methods on the migration behavior, migration kinetics and migration thermodynamics of the liner plasticizer were investigated using the dipping method. In addition, the mechanical properties of the different liners were also investigated.

## 2. Materials and Methods

### 2.1. Materials

Hydroxyl-terminated polybutadiene (HTPB, 99.9%, hydroxyl value ≥ 0.1 mol/100 g) was purchased from Tianjin Science Biochemical Technology Co., Ltd., Tianjin, China. Toluene diisocyanate (TDI, ≥98.0%), dibutyltin dilaurate (DBTL) and triethanolamine (TEA, ≥99.0%) was purchased from Shanghai Aladdin Company, China. Dioctyl sebacate (DOS, ≥97.0%) and octadecylamine (ODA) was purchased from Shanghai Macleans Biochemical Technology Co., Ltd., Shanghai, China. Graphene oxide was prepared using a modified Hummers method.

### 2.2. Preparation of Various Composite Liners

The GO was prepared first, then TDI-modified GO (TGO) and ODA-modified GO (GA) were prepared. The procedure for the preparation of GO and TGO is provided in the Appendix A. Finally, the modified GO/HTPB composite liners were prepared and cured.

The GA was prepared as follows: 0.2 g of prepared GO was weighed and dispersed in 150 mL of deionized water, and the pH of the GO dispersion was adjusted to 8–9. The dispersion was sonicated for 3 h to obtain a homogeneous GO suspension. After the sonication was completed, the mixture was stirred and refluxed at 80 °C. Then 0.3 g of ODA was dissolved in 30 mL of anhydrous ethanol, slowly added to the GO suspension and stirred at 80 °C for 15 h. Finally, the obtained product was filtered and washed with ethanol and dried to obtain GA. The modification mechanism of GA is shown in Figure 1.

The preparation process for each composite liner was as follows: (1) The quantitative GO (0.3 wt%), TGO (0.2 wt%) and GA (0.15 wt%) was placed in acetone, respectively, and then ultrasonicated to obtain a fully dispersed suspension; (2) HTPB and DBTL (1:0.1%) was mixed with each suspension by mechanical stirring at 100 °C to remove acetone and TDI was introduced using the curing coefficient R = 1.3; (3) The mixtures were placed in oil bath at 80 °C and stirred for 2 h, respectively; (4) TEA was added, stirred well and poured into molds (150 × 150 × 2 mm); (5) Each mold was placed into the oven and cured at 80 °C for 36 h to obtain GH, TGH and GAH liner, respectively.

### 2.3. Methods of Characterization Testing

In order to verify the success of GO modification, the components and structures of various GO fillers were tested using Fourier transform infrared (FTIR) and Raman spectroscopy. The surface morphology of GO and modified GO was observed using scanning electron microscopy (SEM). The dispersion of GOs in the matrix was also analyzed by SEM observation of the cross sections of various liners.

The X-ray photoelectron spectroscopy (XPS) was performed using an X-ray photoelectron spectrometer (Shimadzu, KRATOS AXIS SUPRATM) with a monochromated Al Kα X-ray source. Anti-migration tests were conducted using the immersion method. The prepared liners were cut into small pieces measuring 10 × 10 × 2 mm^3^, and their initial mass m0 was recorded. Then, the small pieces were completely immersed in a sealed sample bottle with DOS at 30, 40, 50, 60 and 70 °C. The samples were removed regularly, and the DOS on the surface of the samples was wiped away with filter paper. After being weighed, samples were put back into the sample bottle until the weight of the sample stabilized. During the experiment, the concentration of DOS (*C_t_*) in the sample at time *t* was calculated using the following formula:(1)$$c_{t}=\frac{m_{t}-m_{0}}{m_{t}}$$
where *m_t_* and *m*_0_, respectively, represent the quality of the liner at time *t* and the initial quality of the liner.

Migration coefficient calculation formula:(2)$$\frac{d}{d\sqrt{t}}\left(\frac{\Delta m}{m_{0}}\right)=\frac{2\rho }{\rho_{i}d_{i}}{\left(\frac{D}{\pi }\right)}^{1/2}$$
where ∆*m* represents the amount of DOS migrated into the liner at time *t*, *m_0_* represents the initial mass of the liner, *ρ* represents the density of DOS (0.918 g/cm^3^), *ρ_i_* represents the liner density and *d_i_* represents the thickness of the liner.

The migration activation energy calculation formula is:(3)$$lnD=lnA-\frac{\;E_{a}}{RT}$$
where *D* represents the mobility coefficient, *R* represents the molar constant, *T* represents the thermodynamic temperature, *E_ɑ_* represents the activation energy and *A* represents the frequency factor.

The adhesive properties were tested using an electronic universal tensile testing machine (CMT4254, Shenzhen Xinsansi Material Testing Co., Ltd., Shenzhen, China). The preparation of samples for testing bonding properties is provided in the Appendix A. The tensile rate was 10 mm/min, the ambient temperature was 25 °C, the sample was stretched until it failed and the bond strength was obtained according to the following formula:(4)$$\tau =\frac{F}{A}$$
where *F* represents the maximum load when the bonding surface of the sample was broken and *A* represents the bonding area of the sample.

## 3. Results and Discussion

### 3.1. FTIR Analysis

Figure 2 shows the FTIR spectra of GO, TGO and GA. In the FTIR spectrum of GO, epoxy C-O-C and hydroxyl-OH were observed at 1057 cm^−1^ and 3421 cm^−1^, and C=O on the edge of the carboxyl group appeared at 1728 cm^−1^. After modification by TDI, the reaction of carboxylic acid and the isocyanate groups formed amide groups at the edges of the GO, which led to an increase in the polarity of the GO and a decrease in the intensity of -OH. As a result, the intensity of C=O in the FTIR spectrum of the TGO was enhanced and shifted (1647 cm^−1^), and the N-H bending vibration and the characteristic absorption peak of isocyanate -NCO appeared at 1540 cm^−1^ and 2268 cm^−1^ [29]. Therefore, it can be stated that TDI successfully modified GO. However, after the ODA modification, the FTIR spectrum of GA changed greatly compared to GO. The intensity of the -OH and C=O peaks was significantly weakened, indicating that a large amount of the oxygen-containing functional groups in GO were consumed by the reaction. The C-H stretching vibrations on the methylene and methyl groups appeared at 2919 cm^−1^ and 2849 cm^−1^. The overlapping characteristic peaks of the N-H and SP^2^ carbon structure domains appeared at 1547 cm^−1^. In summary, the ODA was successfully directed to GO.

### 3.2. Raman Spectroscopy

The Raman spectra of GO, TGO and GA are shown in Figure 3. The D peak and G peak are the two main characteristic peaks of carbon materials. In graphene materials, the D peak is the aromatic ring breathing mode activated due to defects, and the G peak is the scattering of the E2 g phonon mode of SP^2^-hybridized carbon atoms [30]. The D and G peaks of GO appeared at 1347 cm^−1^ and 1592 cm^−1^. The G peak of TGO appeared at 1583 cm^−1^ and the G peak of GA appeared at 1589 cm^−1^, both lower than that of GO. This was mainly due to the effect of isolated double bonds and defects [31] and indicated that there were strong interactions between GO, TDI and ODA [32]. The ID/IG is the ratio of the intensities of the two peaks and can be used to measure the degree of chemical reaction. The ID/IG value of GO was 0.86, while TGO and GA were 0.91 and 0.90, respectively, which indicated that GO had more SP^2^ domains and defects after the TDI and ODA modification reactions [33]. In conclusion, it was shown that TDI and ODA successfully reacted with GO.

### 3.3. XPS Analysis

Figure 4 shows the XPS data spectra of GO, TGO and GA. As can be seen in Figure 1a, different bound states appeared in the GO spectrum: C=C (~284.5 eV), C-C (~284.5 eV), C=O (~287.2 eV), C-O-C (~286.6 eV) and C(=O)-O (~288.5 eV) [24,34]. After the ODA modification, a new C-N bonded state appeared in the GA spectrum at ~285.9 eV. The C=C and C-C peaks were enhanced, the C-O-C peak was weakened and the C=O and O-C=O peak positions disappeared. After the modification with TDI, the TGO spectrum showed new C-N bonded states at ~286.1 eV. The bound state -NHCOO (~399.3 eV) indicated the presence of covalent bonds formed by the reaction between TDI and GO. The bound state at -NCO (~401.2 eV) indicated that the GO surface was successfully grafted with isocyanate groups. Because the C/O of TDI was relatively high, the C=C and C-C peaks were therefore enhanced [25]. The C-O and C-O-C peaks were weakened and the C-O-H peak disappeared.

The elemental compositions of GO, TGO and GA are listed in Table 1 and further analyzed. After modification by ODA, the N elemental content of GA reached 1.7%, in addition to a sudden decrease in the elemental content of O from 34.0% to 4.4%. This indicated that the oxygen-containing functional groups were fully reacted and the modified C/O ratio reflected the partial reduction in GO in the reaction. After modification with TDI, the elemental N content of TGO reached 13.7%, while the elemental O content dropped from 34.0% to 14.5%. The -NCO at one end of the TDI reacted with the oxygen-containing functional group, while at the other end -NCO remained on the GO surface; thus, the elemental N content increased more.

In summary, TDI and ODA were successfully modified on GO surfaces.

### 3.4. SEM Analysis

Figure 5 shows the SEM images of GO, TGO and GA. The GO exhibited a more obvious layered structure. Due to the large number of oxygen-containing functional groups, the surface of the GO was rough and had folds (Figure 5a,b). After modification by TDI, the surface of the TGO became uneven with more folds and defects. The original lamellar structure was lost, and the structure became three-dimensional (Figure 5c,d). Moreover, the size of the particles became larger, thus blocking the channels for small molecule migration more effectively when applied to the liner.

After GO was modified by ODA, a large amount of oxygen-containing functional groups were consumed by the reaction. Therefore, compared with GO, GA had a flatter surface and retained a lamellar structure. The size of the lamellae became larger, which also enabled it to effectively inhibit the plasticizer migration (Figure 5e,f). However, the decrease in surface roughness and the decrease in the number of oxygen-containing functional groups accelerated the agglomeration phenomenon; thus, it had some influence on the uniform dispersion of GA in the matrix.

### 3.5. Dispersion Analysis

Equal amounts of GO, TGO, and GA were used to prepare 1 mg/mL of acetone dispersion. The dispersion was observed after ultrasonic dispersion and is shown in Figure 6. The GO dispersion appeared to settle and delaminate after 6 h of standing, and the GA dispersion had completely settled and delaminated, while no obvious particle settling was observed for the TGO dispersion. After 24 h, the suspension of GO also finished settling and stratifying, and the liquid was clarified. In contrast, only a small amount of particle settling occurred in the TGO suspension.

The morphology of the pure HTPB, GH, TGH and GAH liners were observed separately by SEM to determine the dispersion of various GOs in the HTPB matrix. Figure 7 shows the SEM images of the various liners. With the addition of the GOs filler, the cross section of each liner became rough, and it was obvious that the GOs were well dispersed in the matrix. In comparison, the TGO was more uniformly dispersed in the HTPB matrix, while the GA showed some agglomeration. This was mainly due to the large consumption of the oxygen-containing functional groups of GA, which reduced the roughness of the surface and led to a decrease in its ability to inhibit agglomeration.

### 3.6. Migration Resistance Analysis

The migration of the plasticizer components from the composite propellant columns into the liner can seriously affect the stability of the propellant [35], and therefore the migration resistance of propellant liners was investigated and analyzed. In this paper, the migration resistance, migration kinetics and migration thermodynamics of liner layers modified in different ways were analyzed using the solution impregnation method.

Figure 8 shows the trend in the DOS concentration in the liner obtained by impregnation. It was noticed that the addition of the filler significantly improved the migration equilibrium concentration of the plasticizer in the liner, with a significant improvement in the migration resistance. The GO and its modifiers could be used as cross-linking agents in the matrix of the liner. With the addition of fillers, the cross-link density of the liner system increased and the mobility of the molecular chain segments decreased, and thus the channels for constructing the plasticizer molecules were reduced. In addition, the filling of GO and its modifiers led to a decrease in the number of ester groups generated in the liner system and a decrease in the interaction force with the plasticizer molecules, achieving the effect of reducing the migration of plasticizer molecules. Comparing the anti-migration behavior of various filler liners, it was found that the TGH liner had the most superior anti-migration performance. The modification of GO by TDI increased the particle size of the GO. Therefore, after filling to HTPB, the grid voids were blocked more effectively, so that the voids through which the plasticizer molecules could pass were greatly reduced and the anti-migration performance could be achieved on a macroscopic scale. In addition, after the TDI modification, the isocyanate groups were introduced on the surface of the GO, and filled into HTPB in a more complicated mix. The isocyanate group reacted with the hydroxyl group at the end of the HTPB chain to form a block, which made the liner layer more compact. Therefore, the cross-link density of the liner was greatly increased and the free volume inside the liner was further reduced, so that the migration of the plasticizer molecules was effectively hindered. The migration resistance of the GAH liner was improved with the addition of the GA filler; the modification of GO with ODA introduced amide groups with electronegativity in the GAH liner, which formed electrostatic repulsion with the plasticizer-like electron acceptor. At the same time, the more polar groups on the GO surface were also modified to obtain less polar groups, thus reducing the migration of the plasticizer. This reduced the overall polarity of the GAH liner; and the lower the polarity, the better the migration resistance of the liner [36]. This modification method required the introduction of only a trace amount of ODA to significantly improve the migration resistance of the liner.

This research showed that the anti-migration performance of the liner was affected by temperature. The molecular spacing increased with the increase in temperature, and thus the free volume of the liner increased and the plasticizer molecules passed more easily through the migration channel. The higher the temperature, the faster the plasticizer molecules migrated, which in turn increased the plasticizer migration. The migration equilibrium concentration of DOS increased with increasing temperature. By comparison, it was found that the TGH liner was least affected by temperature, and the anti-migration performance of the GAH liner was greatly affected by temperature.

Compared with the pure HTPB liners, the DOS migration concentrations of the three composite liners were significantly reduced: GH (23.28%), TGH (32.25%) and GAH (27.16%). It was observed that the GO/HTPB modified by both methods had a lower DOS migration concentration than the pure HTPB liner and the unmodified GH liner. Compared with the GAH liner, the TGH liner had a greater advantage in anti-migration.

### 3.7. Migration Dynamics Analysis

In addition to the migration concentration, the rate of migration was also an important basis for judging the migration resistance of the liner, and the migration coefficient graphically reflected the migration rate. The migration coefficients were calculated by linearly fitting the point plots using *t*^1/2^ and ∆*m*/*m*_0_ as the horizontal and vertical coordinates, respectively. The results of the linear fit are shown in Figure 9.

The linear correlation coefficients (R2) and slope-calculated diffusion coefficients (DC) of the fitted results are recorded in Table 2. From the figure, it can be found that the migration coefficient of the GO-filled composite liner was significantly lower compared to that of the pure HTPB liner. Among them, the modified GO composite liner reduced the migration coefficient of the pure liner by an order of magnitude. In addition, the migration coefficient of each liner increased with increasing temperature. This can be explained by the thermal movement of molecules: molecules move more vigorously at high temperatures, so the probability of the plasticizer molecules being able to pass through the liner grid gaps in the same time is greater, increasing the plasticizer migration on a macroscopic scale. Comparing the two modified liners, the migration coefficient of the TGH liner was lower than that of the GAH liner, and the migration coefficient of the GAH liner was more affected by temperature. This was because the TDI modified liner had a more stable microstructure with a denser bulk structure that was more difficult for the plasticizer molecules to pass through. The migration resistance of the GAH liner reflected the electrostatic repulsion between the amide groups on the surface of the grafted GO and the electron acceptor (e.g., the plasticizer). As the temperature increased, the plasticizer molecules acquired a higher ability to break through this repulsion. When a large amount of plasticizer entered the liner, the internal structure was disrupted, resulting in a faster migration of the plasticizer molecules.

In summary, the analysis of migration kinetics indicated that the TGH liner had the slowest migration rate of the plasticizer.

### 3.8. Thermodynamic Analysis of Migration

The migration coefficients were plotted against temperature as the horizontal and vertical coordinates in a dotted line plot. Figure 10 shows the results of the linear fit of the DOS migration activation energy, and Table 3 lists the migration activation energies calculated using Equation (3).

Migration activation energy is the most direct reflection of the ease of the molecular migration process. The comparison found that the TGH liner had the best linear correlation (Figure 10c). The migration activation energy of the plasticizer molecules in the TGH liner fluctuated the least with temperature, and the migration activation energy in other liners was greatly affected by temperature. From Table 3, it can be concluded that the migration activation energy of the plasticizer molecules was significantly enhanced by adding GO or its modifiers to HTPB, with the most significant enhancement in the TGH liner, which was 63% higher than that of the pure HTPB liner. This was attributed to the structural stability inside the TGH liner, which made it difficult for the plasticizer to penetrate.

### 3.9. Mechanical Property Analysis

There are many stress processes in the use of propellants, and the liner is prone to interface debonding and accidents. Therefore, the mechanical properties of the liner are also an important evaluation index.

Compared with the pure HTPB liner, the filling of both GO and its modifiers resulted in a significant improvement in the tensile strength and bond strength of the liner (Figure 11). Among them, the GAH liner had the most obvious improvement in its mechanical properties. There were two main reasons for this: First, when GO was modified by ODA, amide groups were formed on its surface, which helped to improve the compatibility of GA with the HTPB matrix, resulting in a more uniform dispersion of GA in the HTPB matrix. Second, the chemical bonds formed by the reaction contributed to the formation of the strong interfacial interactions between them. The mechanical properties of the liner were greatly improved.

The mechanical properties of all three composite liners were superior to those of the pure HTPB liners. Among them, the tensile strength and bond strength of the GAH liner reached 1.12 Mpa and 1.72 Mpa, respectively. Compared with the pure HTPB liner, the tensile strength and bond strength were improved by 522% and 588%. Therefore, the GAH liner had the most significant advantage in terms of mechanical properties.

## 4. Conclusions

In this paper, the modified GO/HTPB composite liner was prepared, and its resistance to plasticizer migration was studied in detail. Compared with the original sample, the migration equilibrium concentrations of the GH, TGH and GAH samples decreased by 23.28%, 32.25% and 27.16%, respectively. In the study of migration dynamics, the linear fitting results showed that the migration coefficient of the modified liner was improved by an order of magnitude. The most significant improvement was observed with the TDI modification. In the study of migration thermodynamics, the migration activation energies of the GH, TGH and GAH samples were increased by 15.13, 17.32 and 11.02 kJ/mol, respectively. In the mechanical properties study, the tensile strengths of the GH, TGH and GAH liners were increased by 50%, 89% and 522%, respectively, and the bond strengths were increased by 280%, 524% and 588%, respectively. We have summarized the properties of the different liners in Appendix A. However, the results of the modification method with the best migration resistance and bonding properties were not obtained in this study. Overall, the structural modification of GO is an effective method for improving the anti-migration and mechanical properties of the liner. It provides a feasible approach to the research on liner structure/function synergistic anti-plasticizer migration.

## Figures and Tables

**Figure 1 polymers-14-05213-f001:**
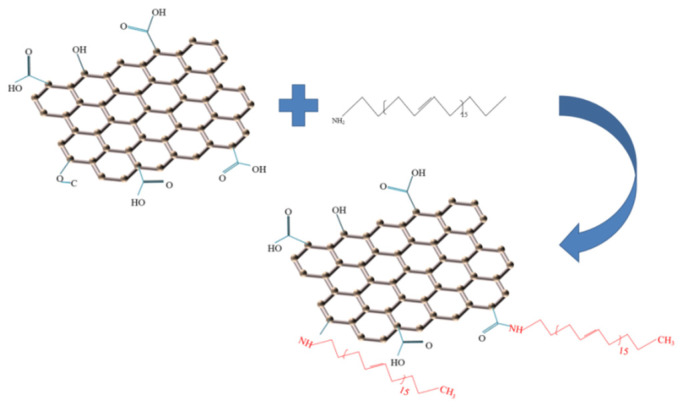
Modification mechanism of GA.

**Figure 2 polymers-14-05213-f002:**
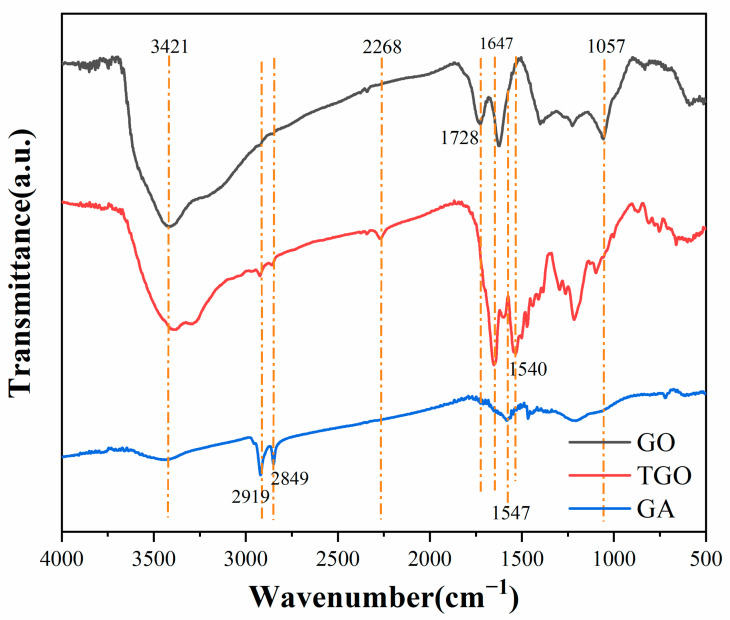
FTIR spectra of GO, TGO and GA.

**Figure 3 polymers-14-05213-f003:**
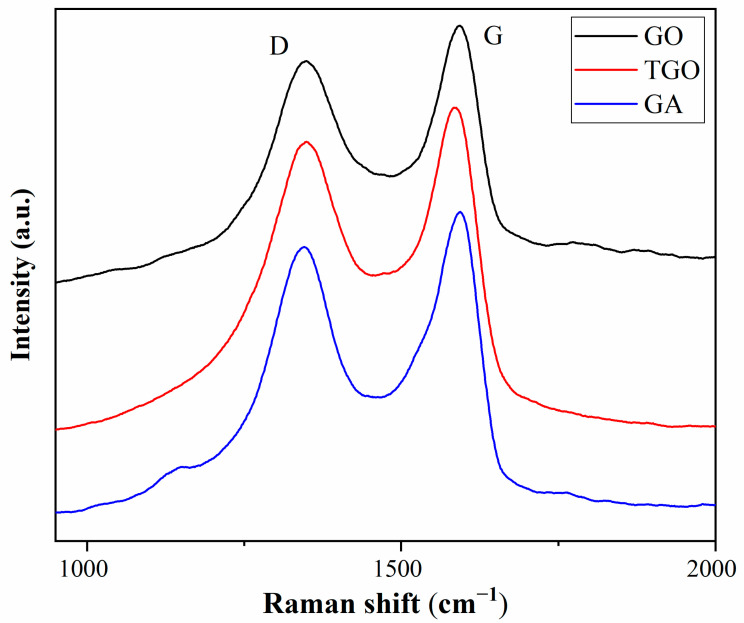
Raman spectra of GO, TGO and GA.

**Figure 4 polymers-14-05213-f004:**
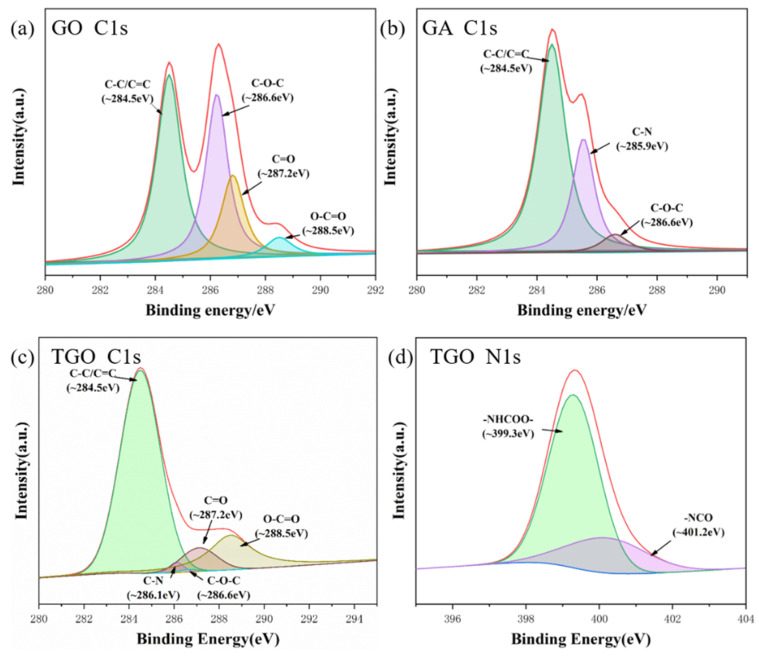
(**a**–**c**) High-resolution C1s XPS spectra of GO, GA and TGO; (**d**) High-resolution N1s XPS spectra of TGO.

**Figure 5 polymers-14-05213-f005:**
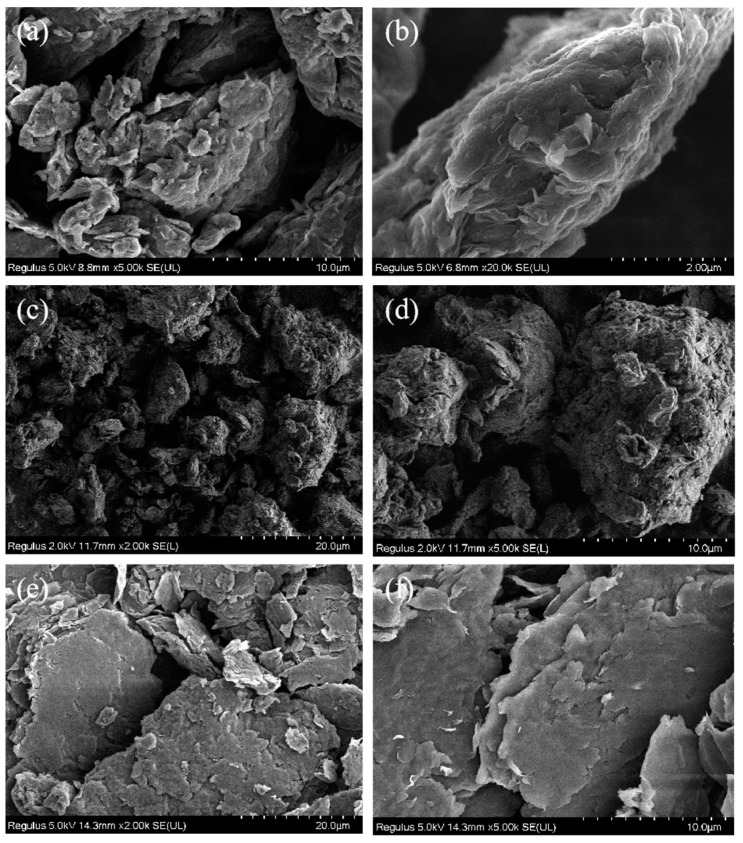
SEM images of GO (**a**,**b**), TGO (**c**,**d**) and GA (**e**,**f**).

**Figure 6 polymers-14-05213-f006:**
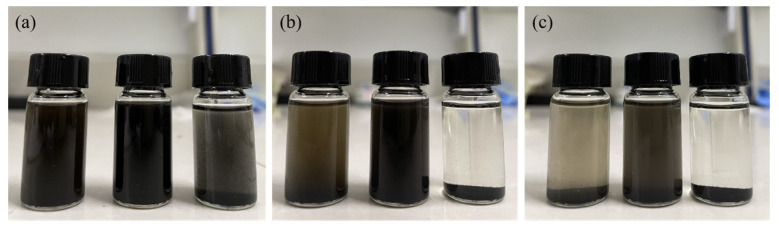
(**a**) Original dispersion; (**b**) Dispersion after 6h standing; (**c**) Dispersion after 24h standing (GO on the left, TGO in the middle, GA on the right).

**Figure 7 polymers-14-05213-f007:**
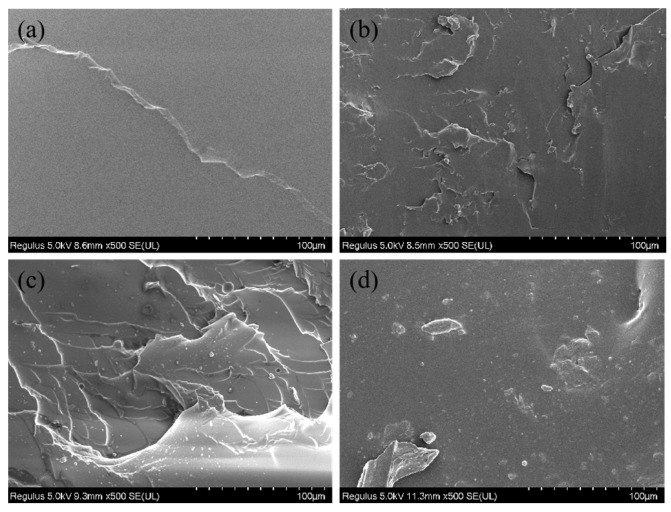
SEM images of the pure HTPB (**a**), GH (**b**), TGH (**c**) and GAH (**d**) liner sections.

**Figure 8 polymers-14-05213-f008:**
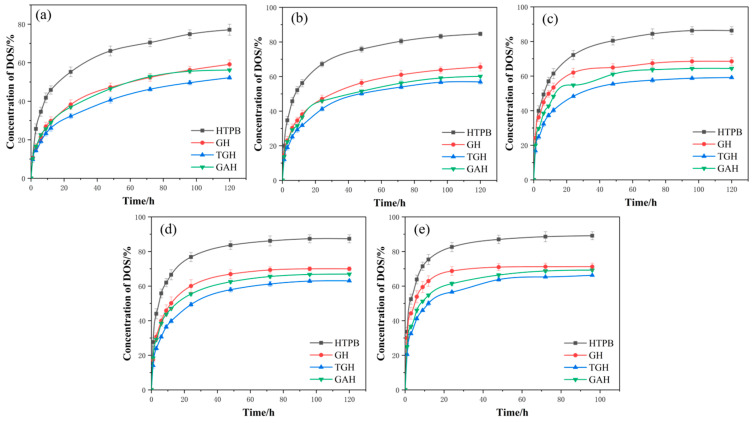
Time-varying DOS concentrations at various temperatures in different liners by the impregnation method. (**a**) 30 °C; (**b**) 40 °C; (**c**) 50 °C; (**d**) 60 °C; (**e**) 70 °C.

**Figure 9 polymers-14-05213-f009:**
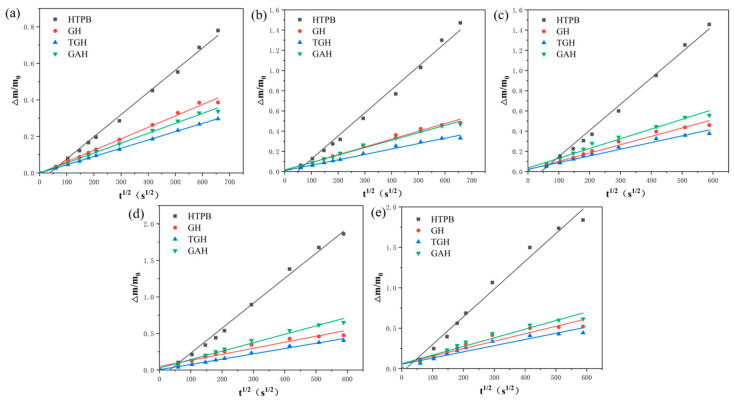
Linear fit of the DOS migration coefficients in the different liners using the dipping method. (**a**) Room temperature; (**b**) 40 °C; (**c**) 50 °C; (**d**) 60 °C; (**e**) 70 °C.

**Figure 10 polymers-14-05213-f010:**
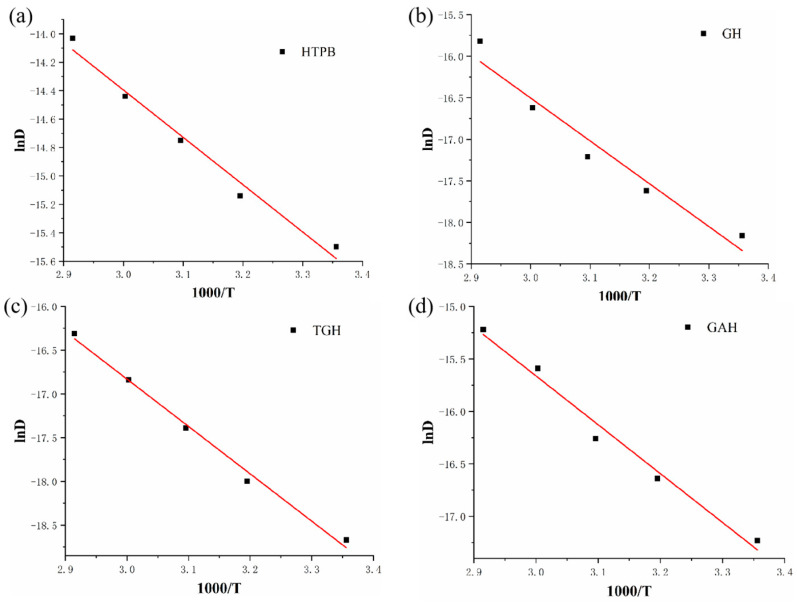
Linear fit of the activation energy of the DOS migration in the different liners by impregnation. (**a**) pure HTPB liner; (**b**) GH liner; (**c**) TGH liner; (**d**) GAH liner.

**Figure 11 polymers-14-05213-f011:**
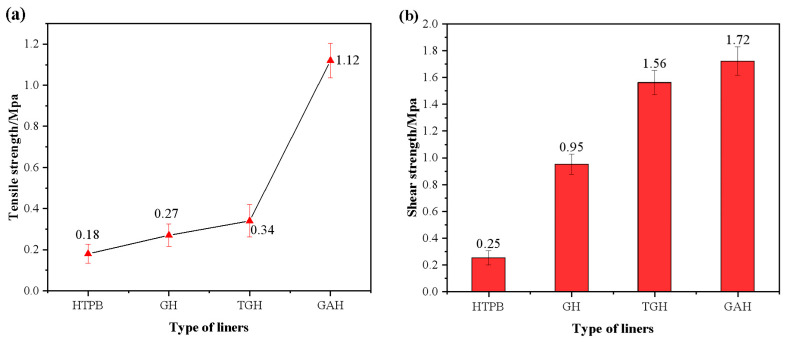
(**a**) Tensile strength and (**b**) Bond strength of the different liners.

**Table 1 polymers-14-05213-t001:** Elemental composition of GO, TGO and GA.

Sample	C/at%	O/at%	N/at%
GO	66.0	34.0	-
TGO	71.7	14.5	13.7
GA	93.9	4.4	1.7

**Table 2 polymers-14-05213-t002:** DOS migration coefficients (m^2^ s^−1^) and linear correlation coefficients in different liners.

	R^2^	30 °C		40 °C		50 °C		60 °C		70 °C	
DC(m^2^ s^−1^)											
HTPB		1.85 × 10^−7^	0.993	2.66 × 10^−7^	0.985	3.93 × 10^−7^	0.994	5.36 × 10^−7^	0.991	8.07 × 10^−7^	0.985
GH		1.30 × 10^−8^	0.992	2.21 × 10^−8^	0.988	3.37 × 10^−8^	0.989	6.05 × 10^−8^	0.941	1.35 × 10^−7^	0.901
TGH		1.10 × 10^−8^	0.999	1.53 × 10^−8^	0.994	2.35 × 10^−8^	0.989	2.81 × 10^−8^	0.996	3.14 × 10^−8^	0.963
GAH		1.99 × 10^−8^	0.995	3.60 × 10^−8^	0.990	6.29 × 10^−8^	0.990	7.76 × 10^−8^	0.983	8.96 × 10^−8^	0.954

**Table 3 polymers-14-05213-t003:** Migration activation energy of DOS in different liner specimens (dipping method).

	E_ɑ_ (kJ/Mol)	r^2^
HTPB	27.69	0.981
GH	42.82	0.951
TGH	45.01	0.992
GAH	38.71	0.982

## Data Availability

Not applicable.

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
