# Peer review of "Improvement in Migration Resistance of Hydroxyl-Terminated Polybutadiene (HTPB) Liners by Using Graphene Barriers"

_polymers, 2022, doi:10.3390/polym14235213_

Round 1

Reviewer 1 Report

The entitled article “Improvement on migration resistance of hydroxyl terminated polybutadiene (HTPB) liners by using Graphene barriers” was carefully reviewed.

It needs revision before consideration for publication in the Polymers.

Limitation of the work, novelty and contributions should be highlighted more.

The introduction still needs to be improved. The following references are relevant to the nanocomposite & their various applications, which should be accommodated in the introduction section to improve the quality of manuscript.

It is better to rewrite properly by referencing the references below.

Polymers 202214(21), 4516. Materials Science for Energy Technologies 4, 92-99, 2021. Materials Science for Energy Technologies 4, 107-112, 2021.

Hydrogen bonding plays a vital role for improving the migration resistance properly. For the confirmation of hydrogen bonding FTIR data can be considered. To confirm the urethane functional groups the following relevant articles can be considered.

Journal of Applied Polymer Science 106 (1), 299-308, 2007.

To improve the quality of the paper SEM/TEM data can be conducted.

More chemistry is required to understand the molecular interaction between polymers.

The provided SEM & TEM images need full scale with high magnification details.

Still in the current state, there are some typographical errors. Therefore, the authors are advised to recheck the whole manuscript.

Overall the article is fairly well written, after addressing the above comments the article may be considered for publication.

Reviewer 2 Report

The present article entitled "Improvement on migration resistance of hydroxyl terminated polybutadiene (HTPB) liners by using Graphene barriers" demonstrates an excellent study carried out with reducing migration of plasticizers by using graphene oxide (GO) with a laminar structure as a filler and can effectively reduce the migration of plasticizers. This is an interesting paper; however, the following comments have to consider:

1. How to select 34% oxygen for GO in your research? The author should try to study the different oxygen content of GO, it can make this research more valuable.

2. The structural characterization of GO used from the Raman measurements must be provided.

3. What is the dispersibility of GO when incorporated into TDI (TGO) and ODA(GA)? The dispersibility greatly influences the composite material, and the authors should add a new chapter to discuss the dispersibility.

4. How to select 0.3 wt% for GO in your research? The author should try to study the different loading of GO, it can make this research more valuable.

5. When the content of GO increases, the composites will get aggregation. How to solve it? The morphology of carbon-based nanomaterials and polymer composites was at nano-/micro-level, so the AFM, SEM and TEM are recommended. How did the authors demonstrate that GOs are uniformly dispersed within the polymer composites?

6. Have the authors considered using graphene nanosheets (GNPs) to increase their migration resistance properties, and if so, why not choose GNPs instead of GO, apart from cost considerations.

7. The authors should compare the composite performance with other similar reported systems in the form of table consisting of important parameters. Please add more studies in new table provided in the text or supplementary.

8. The manuscript should include a more thorough background on related works in literatures. Although the manuscript has cited several papers, it did not cover many references reporting new developments of GO/polymer composites, such as those in Polymers 2022, 14(21), 4560; Journal of Applied Polymer Science, 2022, 139, 31, e52713; Polymers 2022, 14(21), 4551; ACS Applied Materials & Interfaces 2022, 14, 42441–42453; ACS Sustainable Chemistry & Engineering 2022, 10, 23, 7625-7634; Polymers 2022, 14(22), 4796; Journal of Colloid and Interface Science 2021, 599, 611–619; ACS Applied Electronic Materials 2021, 3, 676–686; ACS Applied Materials & Interfaces 2016, 8, 48, 33165-33174.

Overall, the paper shows interesting findings, but major revision should be made before publication.

Round 2

Reviewer 1 Report

The revised article “Improvement on migration resistance of hydroxyl terminated polybutadiene (HTPB) liners by using Graphene barriers” was carefully reviewed.

 The authors have made the suggested corrections carefully.

The paper may be acceptable in the present form.

Reviewer 2 Report

The revised manuscript satisfied my concerns and it is acceptable for publication.